# Acute mental health responses during the COVID-19 pandemic in Australia

Jill M. Newby[1,2]*, Kathleen O'Moore[2], Samantha Tang [2], Helen Christensen[2], Kate Faasse[1]

**1** School of Psychology, UNSW Sydney, Sydney, New South Wales, Australia, **2** Black Dog Institute, UNSW Sydney, Sydney, New South Wales, Australia

* j.newby@unsw.edu.au

**Data Availability Statement:** The data cannot be publicly shared as the dataset contains potentially identifying and sensitive participant information. Data will be made available upon request. For all data requests, please contact the corresponding

## Abstract

The acute and long-term mental health impacts of the COVID-19 pandemic are unknown. The current study examined the acute mental health responses to the COVID-19 pandemic in 5070 adult participants in Australia, using an online survey administered during the peak of the outbreak in Australia (27th March to 7th April 2020). Self-report questionnaires examined COVID-19 fears and behavioural responses to COVID-19, as well as the severity of psychological distress (depression, anxiety and stress), health anxiety, contamination fears, alcohol use, and physical activity. 78% of respondents reported that their mental health had worsened since the outbreak, one quarter (25.9%) were very or extremely worried about contracting COVID-19, and half (52.7%) were worried about family and friends contracting COVID-19. Uncertainty, loneliness and financial worries (50%) were common. Rates of elevated psychological distress were higher than expected, with 62%, 50%, and 64% of respondents reporting elevated depression, anxiety and stress levels respectively, and one in four reporting elevated health anxiety in the past week. Participants with self-reported history of a mental health diagnosis had significantly higher distress, health anxiety, and COVID-19 fears than those without a prior mental health diagnosis. Demographic (e.g., non-binary or different gender identity; Aboriginal and Torres Strait Islander status), occupational (e.g., being a carer or stay at home parent), and psychological (e.g., perceived risk of contracting COVID-19) factors were associated with distress. Results revealed that precautionary behaviours (e.g., washing hands, using hand sanitiser, avoiding social events) were common, although in contrast to previous research, higher engagement in hygiene behaviours was associated with higher stress and anxiety levels. These results highlight the serious acute impact of COVID-19 on the mental health of respondents, and the need for proactive, accessible digital mental health services to address these mental health needs, particularly for those most vulnerable, including people with prior history of mental health problems. Longitudinal research is needed to explore long-term predictors of poor mental health from the COVID-19 pandemic.

author together with the UNSW Human Research Advisory Panel (HREAPC@psy.unsw.edu.au) with the study approval number (3330).

**Funding:** This study was funded by a MRFF Career Development Fellowship to JMN. The funders had no role in study design, data collection and analysis, decision to publish, or preparation of the manuscript.

**Competing interests:** The authors have declared that no competing interests exist.

## Introduction

The novel Coronavirus (COVID-19) first emerged in Wuhan, China in December 2019, and has since evolved into a global pandemic. As of April 27th 2020, there are more than 2.87 million confirmed cases and 198,668 deaths globally with 6,720 confirmed cases, and 83 deaths from COVID-19 in Australia [1]. The COVID-19 pandemic has caused unprecedented disruption to the way most people live, work, study, socialise, and access health care; with widespread travel bans, border closures, lockdowns, social distancing, isolation and quarantine measures enforced by many countries. These changes and their ramifications (e.g., unemployment, social isolation), along with fears of COVID-19 are likely to have significant and long-term impacts on the mental health of the community. Research into past pandemics, such as the 2003 outbreak of Severe Acute Respiratory Syndrome (SARS), has shown higher rates of illness fears, psychological distress (e.g., depression, anxiety, stress), insomnia and other mental health problems (e.g., posttraumatic stress) in people with pre-existing mental illness, front-line health care workers [2], and survivors of severe and life-threatening cases of the disease [3–6].

High quality research into the mental health impacts of COVID-19 is urgently needed [7] to inform evidence-based policy decisions, prevention efforts, treatment programs and community support systems, particularly for those who are most vulnerable and those who are at risk of experiencing poor mental health outcomes during and after this pandemic. In marked contrast to the rapidly growing literature into the physical health consequences of COVID-19, there is currently limited information about the mental health impacts of the COVID-19 outbreak in the general population. However, some recent research has emerged from countries such as China [8–11], Italy [12, 13], India [14, 15], Mexico [16], the United Kingdom [17], USA [18] and Spain [19–21]. In a cross-sectional survey of 52,730 participants in China conducted between the 31st January to the 10th February 2020 [11], 29.3% of respondents experienced mild to moderate psychological distress, and 5.1% experienced severe distress. In another survey of 1210 members of the general public (half of whom were students) conducted between 31st January to 2nd February 2020, Wang et al. [8] found that over half (53.8%) of participants rated the psychological impact of the COVID-19 outbreak as moderate to severe, three quarters were worried about their family members contracting COVID-19, and rates of moderate to severe depression, anxiety and stress were 16.5%, 28.8%, and 8.1% respectively. In a follow-up survey four weeks later, rates of depression, anxiety and stress remained unchanged [22]. In another survey of 7236 self-selected volunteers from 3rd to 17th February 2020, Huang & Zhao [23] found that 20.1%, 35.1%, and 18.2% of respondents reported symptoms of depression, generalised anxiety disorder (GAD), and insomnia on self-report measures. Outside of China, rates of psychological distress have varied across countries and contexts in online mental health surveys, although high rates of psychological distress have been found in countries such as Mexico (50.3% reported psychological distress as moderate to severe) [16], Spain (72% had elevated psychological distress on the GHQ-12) [19], in India (25%, 28%, and 11.6% had moderate to extremely severe depression, anxiety and stress symptoms respectively) [24] during the COVID-19 pandemic.

Together these studies demonstrate the elevated psychological distress in the general community during the initial COVID-19 outbreak. These studies also give some early insights into factors that may increase a person's vulnerability to experiencing poor mental health during the pandemic. Preliminary evidence suggests that i) demographic factors (younger participants, females, college students, and those with low educational attainment) [17, 25], ii) occupational factors (migrant workers, nurses), iii) health-related factors (history of chronic illness, poor self-rated health [8]), and iv) greater exposure to COVID-19 and the worst affected

regions of the outbreak [11], are associated with higher distress levels. In contrast, engaging in precautionary behaviours (e.g., hand hygiene, wearing a mask) have been associated with lower distress [8, 22]. As COVID-19 has spread internationally, more research is urgently needed to explore the mental health impacts of the outbreak, and to identify groups who are vulnerable to poorer mental health in other countries.

To our knowledge there are no published findings on the mental health of the general community during the COVID-19 pandemic in Australia. However, we conducted a previous online survey of the knowledge, attitudes, behaviours and risk perceptions of 2174 people from the general community, shortly after the first death occurred from COVID-19 and when confirmed COVID-19 cases were low in Australia (March 2nd -9th 2020) [26]. In that study, we found one in three participants were very or extremely concerned about an outbreak, and that participants perceived their risk of personally contracting COVID-19 as relatively high (rated as 70% likelihood of contracting the virus). Moreover, most participants (61%) expected that they would experience moderate to severe symptoms of COVID-19 if they contracted the virus. We did not measure mental health outcomes, or how afraid individuals were of personally contracting COVID-19. Therefore, the current study extended our previous survey and investigated the mental health of Australian residents during a 12-day period from 27th March to 7th April 2020, which is now considered to be the time of the peak in new cases, and the steady decline in new cases.

There were several restrictions in place at the time of recruitment, including 1.5 metre social distancing rules [27], and international travel bans. Pubs, hotels, gyms, indoor sporting, cinemas and entertainment venues were closed, restaurants and cafes were restricted to take away or home delivery, religious gatherings and funerals were limited to very small groups (one person per 4 square meter), and there were restrictions to entering aged care homes. As of 25th March 2020 [28], three days prior to recruitment, outdoor events or gatherings were limited to groups of no more than 10 people (with 1.5 metre social distancing). From 28th March 2020, all travellers arriving in Australia from overseas were required to undergo a mandatory 14-day quarantine in designated accommodation. From the 31st March (3 days into recruitment), further restrictions were implemented, limiting movement out of the home, except for shopping for essentials, receiving medical care, undertaking daily exercise or are attending work or school. Gatherings in public were limited to 2 people, except where members of the same household. On the first day (27th March) of the study recruitment period, there was a total of 3378 confirmed cases and 13 deaths related to COVID-19 in Australia, and 328 newly diagnosed cases. Over the next two days, there was an increase of 785 new cases in Australia. Finally, over the remaining days of the study, the number of daily new cases steadily declined, with 93 new cases reported on the last day of recruitment (7th April 2020). There was a total of 5988 confirmed cases (including 3392 active cases) and 49 deaths at the end of the survey period.

Drawing from past research [8, 11, 22] we assessed demographic characteristics, fears of COVID-19, risk perceptions and behavioural responses to the outbreak, psychological distress (depression, anxiety, stress), and alcohol use. We included measures of health anxiety and contamination fears due to their potential role in influencing behaviour, health service use, and anxious reactions to viral outbreaks [29–32]) as well as physical activity levels, and loneliness, due to the expected negative impacts of social distancing measures on these variables, and due to their important role in mental and physical health [33, 34]. Finally, we assessed financial worries, as we expected unemployment, and financial insecurity, which have already resulted from this outbreak, to have significant, negative impacts on mental health [7, 35]. Our primary aim was to provide the first snapshot of the mental health of the general community during the initial COVID-19 outbreak (and enforcement of social distancing laws) in Australia. The

second aim was to explore the relationship between specific demographic and sample characteristics with depression, anxiety and stress, to identify factors that are associated with increased vulnerability for poorer mental health during the COVID-19 pandemic. While we acknowledge that the data from an online survey may not be representative of the entire population, they provide an important opportunity to i) identify vulnerable groups who are at risk of poorer mental health during COVID-19, ii) determine the socio-demographic and psychological factors that predict psychological distress, and iii) examine whether the findings from past pandemics, and from China, apply to the Australian context during the COVID-19 pandemic. Based on research from past pandemics, and Chinese research, we expected that between 20–35% would worry about contracting COVID-19 and experience elevated psychological distress, and that specific demographic variables including younger age, being a student, unemployed, female, or with lower educational attainment would predict higher distress levels in the current cohort. We also expected people with lived experience of prior mental health diagnoses would have higher rates of distress and would be vulnerable to poorer mental health during the current pandemic. Finally, we predicted that engaging in precautionary hygiene behaviours would be associated with lower distress.

## Materials and methods

### Recruitment

Participants were recruited for the online survey via social media posts, with Facebook advertisements targeting all users with i) current country of residence as Australia, and ii) age listed as 18 or above (see S1). Data was collected for 12 days from Friday 27th March to April 7th, 2020. The survey was administered via the Qualtrics survey platform. Each response came from a unique IP address to minimise duplicate entries.

### Ethics approval and consent

The study was approved by the UNSW Human Research Ethics Advisory Panel and the UNSW Human Research Ethics Committee (approval number 3330). All respondents provided electronic informed consent before participating.

### Participants

In total, 5,971 people viewed the participant information page and consent form. Of these, 579 did not complete the consent form, and a further 323 completed only some of the survey questions before discontinuing. This resulted in a final sample of 5071 participants with sufficient data (>70% complete) to include in the analysis. The structured questionnaire took approximately 15 minutes to complete (median time taken: 15.9 minutes).

### Measures

**Demographics.** Information was collected on participants' age group, gender, ethnicity, Aboriginal and Torres Strait Islander status, their highest level of education, carer status (for children, and/or someone with a disability, illness or frail aged) and state of residence within Australia. We also assessed participants' employment status (including whether they had recently lost their job due to COVID-19), the industry of their main job, and the frequency at which they had worked from home during the past week (*not at all*, *a little*, *sometimes*, *most of the time*, *all of the time*).

**General health and mental health.** Participants were asked whether they had a chronic illness (*Yes*, *No*, *Unsure*, *Prefer not to say*), and completed a single-item measure assessing their

*self-rated health* [36], with responses on a 5-point scale from *Poor* to *Excellent*. Participants were asked whether they had ever been diagnosed with a mental health problem such as depression and anxiety (*Yes*, *No*, *Unsure*, *Prefer not to say*), and whether they were currently receiving treatment for a mental health problem including medications, counselling, or psychological therapy (*Yes*, *No*, *Unsure*, *Prefer not to say*).

**Mental health.** Participants were asked to complete single item measures of i) how lonely they were feeling, ii) how worried they were about their financial situation, and iii) how uncertain they were feeling about the future, on a 5-point scale (*not at all*, *a little*, *moderately*, *very*, *extremely*). They were then asked to rate how the COVID-19 outbreak had impacted their mental health. "*Since the COVID-19 outbreak, my mental health has been. . .*", and choose between 5 response options: *A lot worse*, *A little worse*, *Stayed the same*, *A little better*, *A lot better*.

The survey included several validated self-report screening instruments including i) the 21-item Depression Anxiety Stress Scales [37], a validated measure of depression, anxiety and stress symptoms, ii) the Whiteley-6 [38], a brief validated measure of health anxiety severity, iii) the Contamination Obsessions and Washing Compulsions subscale of the revised version of Padua Inventory of Obsessions and Compulsion [39], and iv) a specific measure of behavioural responses to the pandemic based on our prior study [26], and past research investigating behavioural responses to pandemics [40, 41]. Finally, we assessed physical activity levels using the Physical Activity Vital Sign [42] which assessed i) the number of days in the past week they engaged in moderate to strenuous activity, and ii) the average number of minutes they exercised at this level, and screened for hazardous alcohol use using the Modified Alcohol Use Disorders Identification Test (AUDIT-C) [43]. All questionnaire responses were anchored to the past week, except for the AUDIT-C (past month), and the Padua contamination subscale (general). The mental health and lifestyle questionnaires were administered in randomised order to minimise responding biases.

**COVID-19 variables, fears and perceived risk.** Participants were asked about their own COVID-19 status (*I have caught COVID-19 in the past and am now recovered*, *I currently have COVID-19 [confirmed with a diagnostic test]*, *I suspect I have COVID-19*, *I do not have COVID-19 and have not experienced it*, *Unsure*, or *Other (open text)*). They also indicated whether they were in self isolation (*Yes—I am in voluntary self-isolation*, *Yes—I am in forced self-isolation*, *No*). Participants were also asked i) whether any of their family or friends had contracted COVID-19 (*Yes*, *No*, *Unsure*), and ii) how concerned or worried they were that their friends or family members would contract COVID-19 (*not at all*, *a little concerned*, *moderately concerned*, *very concerned*, *extremely concerned*).

Participants were asked five questions relating to their perceived risk from, and worry about, COVID-19. The first question assessed how concerned or worried respondents were about catching COVID-19 on a 5-point scale (*not at all concerned*, *a little concerned*, *moderately concerned*, *very concerned*, *extremely concerned*). They then rated how likely they thought it was that they would catch the virus on a visual analogue scale (VAS) from 0 (*not at all likely*) to 100 (*extremely likely*). They were asked how much they thought they could do personally to protect themselves from catching the virus (perceived behavioural control), on a 0 (*couldn't do anything*) to 100 (*could do a lot*) visual analogue scale. Perceived illness severity was assessed by asking respondents how severe they thought their symptoms would be if they did catch COVID-19 (response options were: *no symptoms*, *mild symptoms*, *moderate symptoms*, *severe symptoms*, *severe symptoms requiring hospitalisation*, and *severe symptoms leading to death*). Finally, participants were asked about how much information they had seen, read or heard about coronavirus (nothing at all, a little, a moderate amount, a lot).

**Health-protective behaviours.**   To assess social distancing, hygiene and buying behaviours, participants were asked whether they had engaged in a total of 16 behaviours during the previous week (see Table 2). Response options for each item were *not at all*, *a little*, *some of the time*, *most of the time*, *all of the time*, and *not applicable*. Items were generated based on our previous study of COVID-19 [26] and from previous research examining health-protective behaviours in response to influenza, SARS and Middle East Respiratory Syndrome (MERS) outbreaks [e.g., 41].

## Statistical analyses

First, we conducted descriptive analyses to describe demographic, sample and clinical characteristics. Second, we conducted chi square analyses (for categorical variables) and independent samples t tests (for dimensional variables) to compare participants with, and without a prior mental health diagnosis, and participants in self-isolation versus those not in self isolation on their questionnaire responses. Third, we conducted separate linear regression analyses to explore the demographic, occupational, and psychological predictors of DASS-21 depression, anxiety and stress severity. We entered demographic predictor variables (gender, age, occupational status, education, Aboriginal and/or Torres Strait Islander and carer status) in the first step. In the second step, we entered general health variables including chronic illness, mental health diagnosis history, and self-rated health. In the third step, we entered uncertainty about the future, loneliness, worry about finances. In the final step, we added COVID-19 variables (whether they were in self-isolation, hygiene behaviours, exposure to COVID-19 information, risk perceptions including perceived likelihood, perceived control, and severity of illness, concern/worry about contracting COVID-19, and concern/worry about loved ones contracting COVID-19. We used a stepwise regression analysis approach to explore the unique variance accounted for by demographic and occupational characteristics, followed by health variables, and then COVID-19-specific variables.

## Results

### Demographics

Demographic characteristics of the sample are depicted in Table 1. Overall, the sample was mostly female (86%), identified as being Caucasian (75%), mainly spoke English at home (91%), and ranged in age from 18 to over 75. Participants were from various states and territories of Australia, with the majority living in the most populated states of New South Wales, Victoria or Queensland. Sixty five percent were working in a paid job, and approximately one third were carers (for children, or people with a disability, illness, or the elderly). Respondents' self-rated health was measured on a scale from poor (1) to excellent (5), with a mean of 3.0 (*SD* = 0.97). The majority of participants rated their health as 'good' (37.7%), 'very good' (24.4%) or 'fair' (24.4%); relatively few participants rated their health as 'poor' (5.3%)' or 'excellent' (5.3%). Seventy percent of respondents reported that they had been diagnosed with a mental health problem such as depression and anxiety in the past, and 45% reported being in current mental health treatment (counselling, medications, therapy).

### Health-related information

Only eight participants (0.2%) reported that they themselves currently have or have had COVID-19, 9.2% were unsure, and 1.2% suspected they had COVID-19. Approximately 4.8% reported their family or friends had caught COVID-19, and 8.2% were unsure. Almost half

**Table 1. Demographic characteristics of the sample.**

| Demographic Variables | N (%) |
|---|---|
| **Gender** | |
| Male | 656 (12.9) |
| Female | 4348 (85.8) |
| Non-binary | 42 (0.8) |
| Different identity | 8 (0.2) |
| Prefer not to say | 15 (0.3) |
| **State** | |
| New South Wales | 1669 (32.9) |
| Victoria | 1236 (24.4) |
| Queensland | 878 (17.3) |
| South Australia | 407 (8.0) |
| Western Australia | 490 (9.77) |
| Tasmania | 215 (4.2) |
| Australian Capital Territory | 141 (2.8) |
| Northern Territory | 31 (0.6) |
| **Age Group** | |
| 18–24 | 268 (5.3) |
| 25–34 | 773 (15.2) |
| 35–44 | 1016 (20.0) |
| 45–54 | 1190 (23.5) |
| 55–64 | 1207 (23.8) |
| 65–74 | 497 (9.8) |
| 75+ | 51 (1.0) |
| Not stated | 67 (1.3) |
| **Ethnicity** | |
| Caucasian (White / European) | 3812 (75.2) |
| Aboriginal and/or Torres Strait Islander | 77 (1.5) |
| Asian | 79 (1.7) |
| Mixed ethnicity or other | 307 (6.1) |
| Prefer not to say or missing | 794 (15.7) |
| **Highest Education** | |
| Less than High school (Year 12 or equivalent) | 275 (5.4) |
| High school only: completed (Year 12) | 419 (8.3) |
| Certificate, or diploma | 1485 (29.3) |
| Bachelor's degree or higher | 2888 (57.0) |
| Not stated | 2 (0.0) |
| **English main language spoken at home** | |
| Yes | 4628 (91.3) |
| **Employment (tick all that apply)** | |
| I am a permanent employee | 2194 (43.3) |
| I am working on a fixed term contract | 362 (7.1) |
| I have a casual job | 432 (8.5) |
| I am self-employed | 388 (7.7) |
| I am an independent contractor | 118 (2.3) |
| I am an at home parent | 221 (4.4) |
| I am a student | 395 (7.8) |
| I am a carer | 129 (2.5) |

(*Continued*)

**Table 1.** (Continued)

| Demographic Variables | N (%) |
|---|---|
| I am retired | 646 (12.7) |
| I am seeking work | 203 (4.0) |
| I am not working and on disability benefits | 258 (5.1) |
| I am not working as I have lost my job due to COVID19 | 314 (6.2) |
| I am not working for other reasons | 341 (6.7) |
| **Industry of main job** | |
| Health care or social assistance | 1039 (32.2) |
| Education and training | 613 (19.0) |
| Administration and social support | 168 (5.5) |
| Professional, scientific and technical services | 242 (7.5) |
| Retail trade | 137 (4.2) |
| Other | 1109 (31.6) |
| **Carer status** | |
| Carer for children | 1196 (23.6) |
| Carer for person with disability, illness or who is frail aged | 772 (15.2) |
| **Isolation** | |
| No | 2475 (48.8) |
| Yes-voluntary self-isolation | 2472 (48.8) |
| Yes–forced self-isolation | 120 (2.4) |
| **COVID-19 diagnosis** | |
| No/Never | 4534 (89.4) |
| Unsure/Other | 462 (9.2) |
| Current diagnosis (confirmed with diagnostic test) | 5 (0.1) |
| Suspect I have COVID-19 | 63 (1.2) |
| I have had COVID-19 in the past and now recovered | 3 (0.1) |
| **Family/friends diagnosed with COVID-19** | |
| Yes | 242 (4.8) |
| No | 4411 (87.0) |
| Unsure | 414 (8.2) |
| **Lifetime mental health diagnosis**[a] | |
| Yes | 3581 (70.8) |
| No | 1351 (26.7) |
| Unsure | 99 (1.9) |
| Prefer not to say | 38 (0.7) |
| **Current mental health treatment** | |
| Yes | 2288 (45.1) |
| No | 2747 (54.2) |
| Unsure | 13 (0.3) |
| Prefer not to say | 21 (0.4) |
| **Current chronic illness**[b] | |
| Yes | 1941 (38.3) |
| No | 2584 (51.0) |
| Unsure | 362 (7.1) |
| Prefer not to say | 34 (0.7) |
| Missing | 148 (2.9) |
| **Self-rated health (in general)**[a] | |
| **Excellent** | 269 (5.3) |

*(Continued)*

**Table 1.** (Continued)

| Demographic Variables | N (%) |
|---|---|
| Very good | 1236 (24.4) |
| Good | 1910 (37.7) |
| Fair | 1235 (24.4) |
| Poor | 270 (5.3) |

[a]: have you ever been diagnosed with a mental health problem such as depression or anxiety?

[b]: do you have a chronic illness?

[c]. $n = 4920$.

(48.8%) reported being in voluntary self-isolation, 2.4% reported being in 'forced self-isolation' and 48.8% were not self-isolating.

## COVID-19 fears and perceived risk

Level of concern and worry about the possibility of contracting COVID-19 was moderate ($M = 2.84$, $SD = 1.07$, range 1–5, where 1 = *not at all*, 5 = *extremely concerned*). A small proportion reported being '*not at all concerned*' (7.6%), 35% reported being '*a little*' concerned, 31.4% were '*moderately concerned*', 17.2% were '*very concerned*', and 8.5% were '*extremely concerned*' about contracting COVID-19. Respondents' ratings of the perceived likelihood of contracting COVID-19 was moderate ($M = 48.25$, $SD = 24.84$; scale from 0 to 100). Perceived behavioural control, or the belief that personal protective behaviours could help prevent infection, had a mean score of 71.64 ($SD = 19.69$). With regard to perceived severity of symptoms if they caught coronavirus, only 0.3% of respondents indicated that they would experience no symptoms; with mild (19.6%) and moderate (43.9%) symptoms most commonly expected. However, one in three respondents perceived the illness severity to be high: with 20.1% indicating they thought they would experience severe symptoms, severe symptoms requiring hospitalisation (12.0%), or severe symptoms leading to death (4.1%). In terms of the amount of information participants had been exposed to about the coronavirus in the past week, most participants (75%) reported having 'a lot' of exposure to information, 21.6% reported a 'moderate amount', whereas very few reported a little (3.3%) or no information at all (0.1%).

## COVID-19 fears (others)

Participants' overall level of concern and worry about friends and loved ones contracting COVID-19 was moderate ($M = 3.53$, $SD = 1.03$, range 1–5, where 1 = *not at all*, 5 = *extremely concerned*). A small proportion reported that they were '*not at all concerned*' (1.6%), 16.5% reported being '*a little*' concerned, 29.2% were '*moderately concerned*', 33.1% were '*very concerned*', and 19.6% '*extremely concerned*' about their friends or family members contracting COVID-19.

## Health-protective behaviours

The percentage of respondents who reported having engaged in a range of distancing and hygiene behaviours during the past week is presented in Table 2. During the previous week, handwashing and social distancing (avoiding social events and gatherings) were the most common behaviours.

**Table 2. Frequency of health-protective behaviours during the past week.**

| | N/A | Not at all | A little | Some of the time | Most of the time | All of the time |
|---|---|---|---|---|---|---|
| **Avoided going to work or university** | 1702 (33.58) | 1120 (22.10) | 170 (3.35) | 197 (3.89) | 306 (6.04) | 1567 (30.91) |
| **Avoided using public transport** | 1828 (36.06) | 142 (2.80) | 70 (1.38) | 75 (1.48) | 199 (3.93) | 2748 (54.21) |
| **Avoided flying domestically or internationally** | 2323 (45.83) | 113 (2.23) | 22 (0.43) | 19 (0.37) | 34 (0.67) | 2549 (50.29) |
| **Avoided social events or public gatherings** | 234 (4.62) | 47 (0.93) | 58 (1.14) | 62 (1.22) | 492 (9.71) | 4168 (82.23) |
| **Avoiding socialising (in person) with anyone outside of your household** | 82 (1.62) | 90 (1.78) | 170 (3.35) | 225 (4.44) | 1495 (29.49) | 2997 (59.12) |
| **Avoided going to hospitals or going to the doctor unless absolutely necessary** | 1015 (20.02) | 280 (5.52) | 167 (3.29) | 155 (3.06) | 561 (11.07) | 2881 (56.84) |
| **Avoided going into shops** | 35 (0.69) | 275 (5.43) | 493 (9.73) | 1017 (20.06) | 2533 (49.97) | 706 (13.93) |
| **Avoided staying in hotels, hostels, or Airbnb's** | 2572 (50.74) | 108 (2.13) | 13 (0.26) | 14 (0.28) | 37 (0.73) | 2315 (45.67) |
| **Avoided sending your children to school or childcare** | 3745 (73.88) | 217 (4.28) | 42 (0.83) | 67 (1.32) | 123 (2.43) | 865 (17.06) |
| **Stayed at home as much as possible** | 38 (0.75) | 31 (0.61) | 56 (1.10) | 219 (4.32) | 2310 (45.57) | 2406 (47.46) |
| **Cleaned or disinfected things you touch (such as doorknobs or hard surfaces)** | 31 (0.61) | 592 (11.68) | 697 (13.75) | 1387 (27.36) | 1390 (27.42) | 964 (19.02) |
| **Used sanitising hand gel to clean your hands** | 92 (1.81) | 441 (8.70) | 428 (8.44) | 1153 (22.75) | 1286 (25.37) | 1661 (32.77) |
| **Washed your hands thoroughly** | 10 (0.20) | 7 (0.14) | 34 (0.67) | 150 (2.96) | 1382 (27.26) | 3475 (68.55) |
| **Worn a face mask when going out in public** | 261 (5.15) | 4067 (80.23) | 193 (3.81) | 223 (4.40) | 148 (2.92) | 169 (3.33) |
| **Avoided touching objects or surfaces knowing they have been touched by other people** | 77 (1.52) | 188 (3.71) | 416 (8.21) | 881 (17.38) | 2005 (39.55) | 1493 (29.45) |
| **Purchased significantly more than you normally would when grocery shopping** | 73 (1.44) | 2008 (39.61) | 1406 (27.74) | 927 (18.29) | 398 (7.85) | 248 (4.89) |

Numbers represent n and proportion (%) in brackets.

## Mental health

More than three quarters of participants reported that their mental health had been worse since the outbreak, with 55.1% selecting '*a little worse*', and 22.9% selecting '*a lot worse*'. A small proportion reported improvements in their mental health since the outbreak (5.5%) (see Fig 1). A chi square analysis revealed that there was a significant difference in the impact of COVID-19 on mental health for participants with and without a prior mental health diagnosis ($\chi^2$ (4) = 141.44, p < .001), with 26.6% of those with a prior mental health diagnosis saying their mental health had been 'a lot worse', relative to 13.4% in the group without a mental health diagnosis.

Almost 80% of individuals reported moderate to extreme levels of uncertainty about the future; half (50.1%) reported feeling moderately to extremely lonely, and half reported moderate to extreme worry about their financial situation (50.1%). See Fig 2 for results.

Table 3 shows the proportion of participants who scored across the severity categories of the DASS-21 subscales. Only 38.2% of respondents scored in the normal range for depression, 50.2% in the normal range for anxiety, and 45.5% for stress. In contrast, 37.1%, 29.1%, and 33.6% fell in the mild to moderate range for depression, anxiety, and stress respectively, whereas 24.1%, 20.3%, and 20.4% reported severe or extremely severe stress levels. On the Whiteley-6, 21.6% scored in the range indicating elevated health anxiety. Of the participants

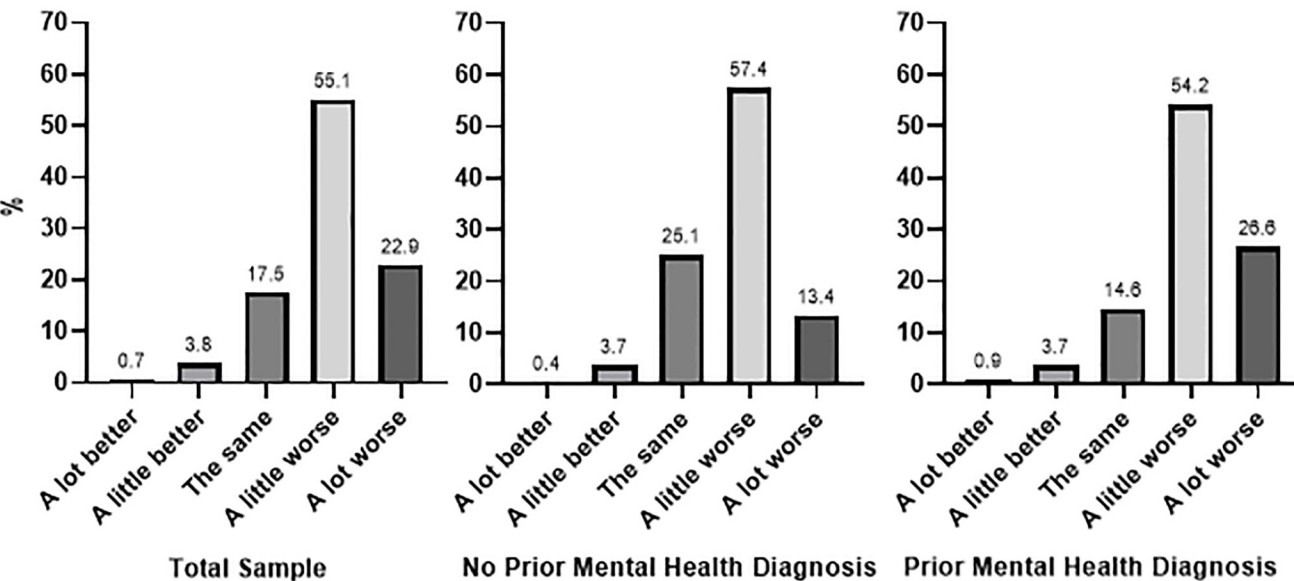

**Fig 1. Proportion of participants reporting how their mental health has been since the start of the COVID-19 outbreak, in the total sample (Left), the sub-sample with a prior mental health diagnosis (middle) and no prior mental health diagnosis (right).**

who had valid scores on the Physical Activity Vital Sign (N = 4845), 42.7% met national guidelines for 150 minutes of moderate to vigorous physical activity in the past week. On the AUDIT-C brief screener for alcohol use, approximately 52.7% showed hazardous drinking levels. Hazardous drinking levels were defined as an AUDIT-C score of 3 or more for women and other genders, and 4 or more for men [43, 44].

**Impact of prior mental health diagnosis.** People with and without a self-reported history of mental health diagnosis were compared in their severity of COVID-19 fears, mental health, distress, health anxiety, alcohol use, contamination fears, and physical activity. People with a

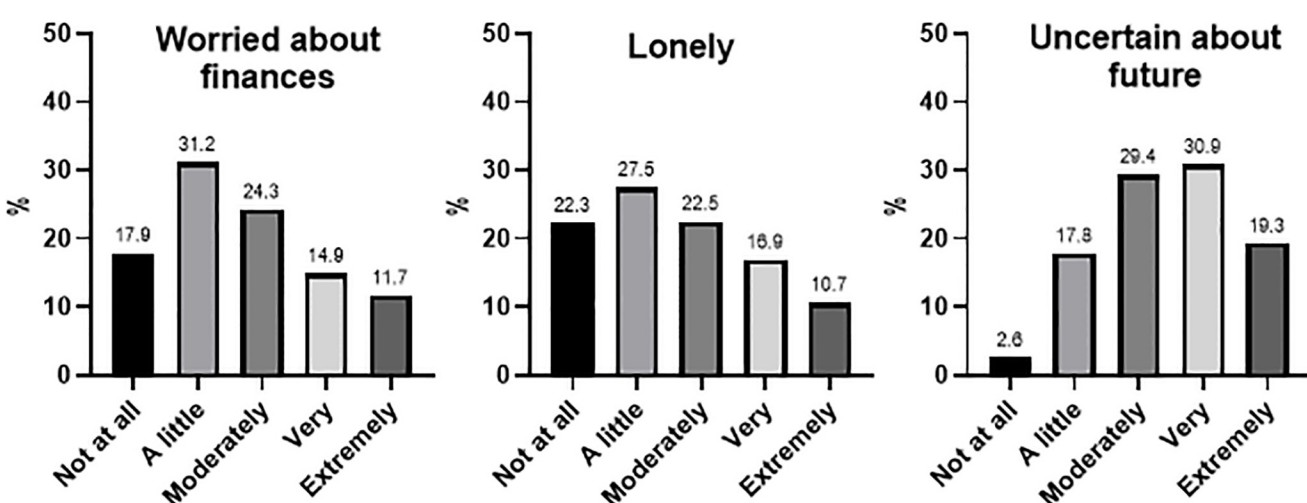

**Fig 2. Proportion (% of total sample) of participants reporting worry about finances, uncertainty about the future and feelings of loneliness.**

**Table 3. Psychological distress, health anxiety, alcohol use, and physical activity.**

|  | Normal | Mild | Moderate | Severe | Extremely Severe |
|---|---|---|---|---|---|
| **DASS-21** | **n (%)** | **n (%)** | **n (%)** | **n (%)** | **n (%)** |
| **Depression Subscale** | 1936 (38.19) | 765 (15.09) | 1124 (22.17) | 533 (10.51) | 691 (13.63) |
| **Anxiety Subscale** | 2546 (50.23) | 434 (8.56) | 1039 (20.50) | 397 (7.83) | 633 (12.49) |
| **Stress Subscale** | 2308 (45.53) | 778 (15.35) | 927 (18.29) | 720 (14.20) | 316 (6.23) |
|  | **M** | **SD** | **Median** |  |  |
| **DASS-21 Total** | 40.19 | 25.07 | 36.00 |  |  |
| **DASS-21 Depression Subscale** | 14.14 | 10.56 | 12.00 |  |  |
| **DASS-21 Anxiety Subscale** | 8.98 | 8.21 | 6.00 |  |  |
| **DASS-21 Stress Subscale** | 17.07 | 9.49 | 16.00 |  |  |
| **Whiteley-6 Total (Health Anxiety)** | 13.18 | 5.61 | 12.00 |  |  |
| **Padua Contamination & Washing Subscale** [a] | 10.76 | 8.78 | 8.00 |  |  |
| **Physical activity vital sign** [b] | 186.86 | 369.39 | 120.00 |  |  |
| **AUDIT-C (alcohol)** [c] | 3.66 | 2.02 | 3.00 |  |  |

DASS-21 = Depression Anxiety Stress 21-item scale.

[a]. n = 4928,

[b]. n = 4845.

[c]. n = 4828

previous self-reported mental health diagnosis reported higher uncertainty, loneliness, financial worries, COVID-19 fears (self and others), believed they were more likely to contract COVID-19, had lower perceived behavioural control, had higher rates of psychological distress, health anxiety and contamination fears, and lower physical activity than those without a self-reported mental health diagnosis history. There were no differences in alcohol use between these groups (see S1 Table).

**Impact of self-isolation.** Compared to people who were not in self isolation, people who self-reported being in self-isolation reported higher uncertainty, loneliness, financial worries, and COVID-19 fears (self and others), rated the symptoms of COVID-19 as more serious, but believed they were less likely to contract COVID-19, and perceived more behavioural control over COVID-19. They also had higher rates of psychological distress, health anxiety and contamination fears, and lower alcohol use than those not in isolation. There were no differences in physical activity between these groups (see S2 Table).

**Predictors of depression, anxiety and stress.** The final linear regression models are presented in Table 4.

*Depression.* Demographic variables accounted for 10.8% of the variance ($R^2_{change}$ = 0.11, SE = 10.02, $F_{change}$ (18, 4971), = 33.32, p < .001). Entering the mental health diagnosis, chronic illness, and self-rated health variables accounted for 9.5% of additional variance ($R^2_{change}$ = 0.095, SE = 9.47, $F_{change}$ (3, 4788), = 191.73, p < .001). In the third step, entering mental health variables accounted for 27.5% unique variance ($R^2_{change}$ = 0.28, SE = 7.66, $F_{change}$ (3, 4785), = 845.35, p < .001). Finally, the COVID-19 variables accounted for 0.7% unique variance ($R^2_{change}$ = 0.007, SE = 7.61, $F_{change}$ (3, 4777), = 8.02, p < .001). The final model is presented in Table 4 and accounted for 48.5% of the variance in depression scores.

Controlling for the other variables in the model, being female, more well educated, older, and having better self-rated health were all associated with lower depression, whereas being a student, retired, carer or stay at home parent were associated with higher depression. Mental health and chronic illness diagnoses were associated with higher depression, as were increased

**Table 4. Predictors of depression, anxiety and stress severity (DASS-21 scores).**

| Variable | DASS-21 Depression | | | | | DASS-21 Anxiety | | | | | DASS-21 Stress | | | | |
|---|---|---|---|---|---|---|---|---|---|---|---|---|---|---|---|
| | B | SE | Exp(B) | t | p | B | SE | Exp(B) | t | p | B | SE | Exp(B) | t | p |
| Constant | 5.51 | 1.43 | | 3.84 | 0.00 | 1.05 | 1.23 | | 0.85 | 0.39 | 3.87 | 1.40 | | 2.76 | 0.01 |
| **Gender** | | | | | | | | | | | | | | | |
| Male (RC) | | | | | | | | | | | | | | | |
| Female | -1.08 | 0.33 | -0.04 | -3.27 | **0.00** | 0.60 | 0.28 | 0.03 | 2.10 | **0.04** | 0.36 | 0.32 | 0.01 | 1.11 | 0.27 |
| Non-binary or different identity | 0.57 | 1.16 | 0.01 | 0.49 | 0.62 | 1.71 | 1.00 | 0.02 | 1.71 | **0.09** | 3.69 | 1.14 | 0.04 | 3.25 | **0.00** |
| Prefer not to say | -0.68 | 2.33 | 0.00 | -0.29 | 0.77 | 4.60 | 2.00 | 0.03 | 2.30 | **0.02** | 3.42 | 2.27 | 0.02 | 1.50 | 0.13 |
| **Age** | | | | | | | | | | | | | | | |
| 18 to 24 (RC) | | | | | | | | | | | | | | | |
| 25–34 | -1.84 | 0.58 | -0.06 | -3.16 | **0.00** | -2.17 | 0.50 | -0.10 | -4.34 | **0.00** | -1.58 | 0.57 | -0.06 | -2.77 | **0.01** |
| 35–44 | -2.39 | 0.58 | -0.09 | -4.12 | **0.00** | -3.21 | 0.50 | -0.16 | -6.46 | **0.00** | -1.69 | 0.57 | -0.07 | -2.98 | **0.00** |
| 45–54 | -2.33 | 0.58 | -0.09 | -4.02 | **0.00** | -4.06 | 0.50 | -0.21 | -8.16 | **0.00** | -3.08 | 0.57 | -0.14 | -5.43 | **0.00** |
| 55–64 | -2.34 | 0.59 | -0.09 | -3.98 | **0.00** | -4.66 | 0.51 | -0.24 | -9.22 | **0.00** | -4.47 | 0.57 | -0.20 | -7.77 | **0.00** |
| 65–74 | -3.27 | 0.73 | -0.09 | -4.50 | **0.00** | -5.41 | 0.62 | -0.20 | -8.67 | **0.00** | -6.03 | 0.71 | -0.19 | -8.48 | **0.00** |
| 75 and older | -3.46 | 1.30 | -0.03 | -2.66 | **0.01** | -4.82 | 1.12 | -0.06 | -4.31 | **0.00** | -6.63 | 1.27 | -0.07 | -5.22 | **0.00** |
| **Aboriginal and/or Torres Strait Islander** | 1.46 | 0.90 | 0.02 | 1.62 | 0.11 | 1.63 | 0.77 | 0.02 | 2.11 | **0.04** | 1.94 | 0.88 | 0.02 | 2.21 | **0.03** |
| **Education** | | | | | | | | | | | | | | | |
| Less than high school (RC) | | | | | | | | | | | | | | | |
| High school only | 0.08 | 0.62 | 0.00 | 0.13 | 0.90 | -0.75 | 0.53 | -0.02 | -1.41 | 0.16 | -0.70 | 0.61 | -0.02 | -1.15 | 0.25 |
| Trade certificate or diploma | -0.90 | 0.52 | -0.04 | -1.74 | **0.08** | -0.98 | 0.44 | -0.05 | -2.20 | **0.03** | -0.84 | 0.51 | -0.04 | -1.67 | **0.09** |
| Bachelor's degree or higher | -1.46 | 0.51 | -0.07 | -2.87 | **0.00** | -1.81 | 0.44 | -0.11 | -4.16 | **0.00** | -0.71 | 0.50 | -0.04 | -1.43 | 0.15 |
| **Employment Status** | | | | | | | | | | | | | | | |
| Paid employment (RC) | | | | | | | | | | | | | | | |
| Unemployed | 0.04 | 0.55 | 0.00 | 0.07 | 0.94 | -0.41 | 0.47 | -0.01 | -0.88 | 0.38 | -0.68 | 0.54 | -0.02 | -1.26 | 0.21 |
| Student | 2.26 | 0.32 | 0.08 | 7.17 | **0.00** | 1.08 | 0.27 | 0.05 | 4.00 | **0.00** | 0.15 | 0.31 | 0.01 | 0.49 | 0.63 |
| Retired | 0.82 | 0.47 | 0.03 | 1.74 | **0.08** | 0.19 | 0.41 | 0.01 | 0.47 | 0.63 | -0.23 | 0.46 | -0.01 | -0.50 | 0.62 |
| At home parent | 1.01 | 0.57 | 0.02 | 1.77 | **0.08** | -0.34 | 0.49 | -0.01 | -0.69 | 0.49 | 1.22 | 0.56 | 0.03 | 2.19 | **0.03** |
| Carer | 1.54 | 0.71 | 0.02 | 2.18 | **0.03** | 0.36 | 0.61 | 0.01 | 0.59 | 0.56 | 0.59 | 0.69 | 0.01 | 0.85 | 0.39 |
| **Chronic illness** | 0.33 | 0.19 | 0.02 | 1.72 | **0.08** | 0.57 | 0.17 | 0.04 | 3.44 | **0.00** | 0.38 | 0.19 | 0.03 | 2.01 | **0.04** |
| **Mental health diagnosis** | 2.23 | 0.24 | 0.10 | 9.38 | **0.00** | 1.88 | 0.20 | 0.11 | 9.22 | **0.00** | 2.51 | 0.23 | 0.13 | 10.81 | **0.00** |
| **Self-rated health** | -1.40 | 0.13 | -0.13 | -10.51 | **0.00** | -0.83 | 0.11 | -0.10 | -7.25 | **0.00** | -0.63 | 0.13 | -0.06 | -4.81 | **0.00** |
| **Uncertainty about future** | 2.07 | 0.13 | 0.21 | 15.75 | **0.00** | 1.26 | 0.11 | 0.16 | 11.17 | **0.00** | 1.96 | 0.13 | 0.22 | 15.24 | **0.00** |
| **Loneliness** | 3.24 | 0.10 | 0.39 | 32.37 | **0.00** | 1.38 | 0.09 | 0.22 | 16.09 | **0.00** | 1.82 | 0.10 | 0.25 | 18.64 | **0.00** |
| **Worry about finances** | 0.73 | 0.10 | 0.09 | 7.04 | **0.00** | 0.46 | 0.09 | 0.07 | 5.19 | **0.00** | 0.40 | 0.10 | 0.05 | 3.95 | **0.00** |
| **Self-isolation** | -0.05 | 0.23 | 0.00 | -0.23 | 0.82 | 0.33 | 0.20 | 0.02 | 1.66 | 0.10 | -0.11 | 0.23 | -0.01 | -0.50 | 0.62 |
| **Hygiene behaviours** | -0.08 | 0.05 | -0.02 | -1.67 | 0.10 | 0.28 | 0.04 | 0.08 | 6.73 | **0.00** | 0.17 | 0.05 | 0.04 | 3.57 | **0.00** |
| **Exposure to COVID-19 information** | 0.13 | 0.21 | 0.01 | 0.61 | 0.54 | -0.58 | 0.18 | -0.04 | -3.16 | **0.00** | -0.09 | 0.21 | 0.00 | -0.43 | 0.67 |
| **Concern/worry about contracting COVID-19** | -0.53 | 0.15 | -0.05 | -3.68 | **0.00** | 0.47 | 0.12 | 0.06 | 3.75 | **0.00** | 0.20 | 0.14 | 0.02 | 1.39 | 0.17 |
| **Likelihood of contracting COVID-19** | 0.01 | 0.01 | 0.03 | 2.15 | **0.03** | 0.00 | 0.00 | 0.01 | 1.00 | 0.32 | 0.01 | 0.01 | 0.03 | 2.48 | **0.01** |
| **Perceived control** | -0.04 | 0.01 | -0.07 | -5.94 | **0.00** | -0.02 | 0.01 | -0.05 | -3.89 | **0.00** | -0.02 | 0.01 | -0.05 | -3.95 | **0.00** |
| **Severity of illness** | 0.26 | 0.13 | 0.03 | 2.02 | **0.04** | 0.30 | 0.11 | 0.04 | 2.67 | **0.01** | -0.02 | 0.13 | 0.00 | -0.14 | 0.89 |
| **Concern/worry about loved ones contracting COVID-19** | 0.01 | 0.13 | 0.00 | 0.04 | 0.97 | 0.37 | 0.11 | 0.05 | 3.30 | **0.00** | 0.75 | 0.13 | 0.08 | 5.84 | **0.00** |

B: N = 4810. Unstandardized coefficient; SE: Standard error; Exp(B): Exponentiated regression coefficient; RC: Reference category.

uncertainty about the future, loneliness, and financial worries. Of the COVID-19 variables, higher worry about COVID-19 and perceived behavioural control over COVID-19 infection were associated with lower depression, whereas perceiving higher illness severity was associated with higher depression.

*Anxiety*. In the first step, demographic variables accounted for 10.7% of the variance in anxiety scores ($R^2_{change}$ = 0.11, SE = 7.77, $F_{change}$ (18, 4791), = 33.05, p < .001). Entering the health variables (mental health diagnosis, chronic illness, and self-rated health) accounted for 8.3% of additional variance ($R^2_{change}$ = 0.083, SE = 7.40, $F_{change}$ (3, 4788), = 163.28, p < .001). In the third step, entering mental health variables accounted for 15.3% unique variance ($R^2_{change}$ = 0.15, SE = 6.67, $F_{change}$ (3, 4785), = 372.11, p < .001). Finally, the COVID-19 variables accounted for 2.7% unique variance ($R^2_{change}$ = 0.027, SE = 6.53, $F_{change}$ (3, 4777), = 25.55, p < .001). The final model is presented in Table 4 and accounted for 36.5% of the variance in anxiety scores.

Controlling for other variables in the model, being female, non-binary or different gender identity, and being Aboriginal and/or Torres Strait Islander were predictors of higher anxiety. Older age, and more well educated (certificate, degree or higher) were predictors of lower anxiety. In contrast to depression, only being a student predicted worse anxiety. Having a chronic illness, and prior history of mental health diagnosis were associated with higher anxiety, whereas better self-rated health was a predictor of lower anxiety. Similar to depression, increased uncertainty about the future, loneliness, and financial worries were also associated with higher anxiety. Of the COVID-19 variables, more hygiene behaviours, worry about COVID-19, worry about loved ones contracting COVID-19, and higher perceived illness severity were predictors of higher anxiety, whereas increased exposure to COVID-19 information, and perceived control over COVID-19 predicted lower anxiety.

*Stress*. In the first step, demographic variables accounted for 10.8% of the variance in anxiety scores ($R^2_{change}$ = 0.11, SE = 8.99, $F_{change}$ (18, 4791), = 33.49, p < .001). Entering the health variables (mental health diagnosis, chronic illness, and self-rated health) accounted for 6.9% of additional variance ($R^2_{change}$ = 0.069, SE = 8.63, $F_{change}$ (3, 4788), = 135.07, p < .001). In the third step, entering mental health variables accounted for 19.4% unique variance ($R^2_{change}$ = 0.19, SE = 7.54, $F_{change}$ (3, 4785), = 496.74, p < .001). Finally, the COVID-19 variables accounted for 1.8% unique variance ($R^2_{change}$ = 0.018, SE = 7.44, $F_{change}$ (3, 4777), = 17.68, p < .001). The final model is presented in Table 4 and accounted for 38.9% of the variance in stress scores.

Controlling for other variables in the model, identifying as non-binary or different gender identity, Aboriginal and/or Torres Strait Islander, predicted higher stress. Being more well-educated with a trade certificate, and older age, were predictors of lower stress. Being a stay at home parent was a predictor of higher stress. Having a chronic illness, and prior history of mental health diagnosis were associated with higher stress, whereas better self-rated health was a predictor of lower stress. Increased uncertainty about the future, loneliness, and financial worries were also associated with higher stress. Of the COVID-19 variables, more hygiene behaviours, worry about loved ones contracting COVID-19, and higher perceived likelihood of contacting COVID 19 were predictors of higher stress. Higher perceived control over COVID-19 predicted lower stress.

## Discussion

This survey presents the first insight into how the COVID-19 pandemic has impacted the mental health of people living in Australia, in a sample of 5070 individuals. Rapidly disseminating an online survey enabled us to assess a large number of participants during the peak of

the pandemic in Australia to identify fears and acute distress and identify the relationship between demographic and psychological predictors of mental health. While very few individuals reported that they (0.15%) or their family/friends (4.8%) had contracted COVID-19, one quarter (25.9%) of respondents were very or extremely worried about contracting COVID-19, and over half (52.7%) were very or extremely worried about their family and friends contracting COVID-19. Almost four in five participants reported that since the outbreak their mental health had worsened, with over half (55%) saying it had worsened a little, and almost a quarter of respondents (23%) saying it had worsened a lot. A small minority reported better mental health (4.8%). Results showed that many people are experiencing high levels of uncertainty about the future (80%), and half of respondents reporting moderate to extreme loneliness and worry about their financial situation. Given loneliness, social isolation, and financial stress are significant risk factors for poor mental and physical health, and risk factors for suicidal ideation [e.g., 33, 34, 45], these findings are concerning.

To rapidly respond to the evolving COVID-19 situation, we administered online validated self-report questionnaires rather than diagnostic interviews. It is important to note that these questionnaires assessed *symptoms* of distress during the past week and should not be taken as indicative of a 'diagnosis' of a depressive or anxiety disorder, and that the sample was comprised of mostly women and a high proportion of individuals with lived experience of mental health problems, and so is unlikely to be representative of the broader Australian general community. We found higher than expected levels of acute distress based on research in China during the COVID-19 pandemic [8], and compared to normative data [37, 46]. Between 20.3–24.1% of the current sample were experiencing severe or extremely severe levels of depression, anxiety and stress, and a further 18–22% moderate symptoms. Only 38% of the current sample had normal depression, 50% had normal anxiety, and 46% had normal stress levels, whereas in the Chinese sample reported by Wang et al. [8] 64–69% had normal anxiety, stress and depression on the DASS-21. Estimates of the rates of psychological distress in other countries, such as Spain, India and Italy have varied widely, but range from one quarter [24] to 70% of participants [19]. The differences between our results and past studies may be due to the high proportion of people with pre-existing mental health diagnoses (70%) in our sample, which have been shown to be a vulnerable group [8, 11], or because of the significant proportion with a self-reported chronic illness (38%), who may be more susceptible to more severe COVID-19 disease, and therefore more distressed. Having a personal history of chronic illness predicted of higher depression, anxiety and stress, whereas better self-rated health was associated with better mental health. Compared to the Australian population, this sample appeared to have poorer health, with 30% reported being in fair or poor health (compared to 15% in the Australian population), and 30% reporting being in very good or excellent health (compared to 56% of Australians) [47].

Our data gave some insights into other demographic variables which were associated with higher psychological distress. Specific occupational factors predicted higher distress levels: student status (depression and anxiety), being an at home parent (depression and stress), or being a carer or retired (predicted higher depression). Education was associated with lower psychological distress. In contrast to past research, identifying as female predicted lower depression, however identifying as non-binary or a different gender identity was associated with higher self-reported anxiety and stress. Identifying as Aboriginal or Torres Strait Islander also predicted worse anxiety and stress levels. There was a relatively small number of Indigenous Australians and respondents who identified as non-binary or different gender identity, in the sample. However, this preliminary evidence suggests these groups may be vulnerable to poorer mental health during the current pandemic. Longitudinal research is needed to explore the longer term predictors of poorer mental health over time.

Our results confirm fears about the potential impact of the COVID-19 pandemic on people with lived experience of mental illness [7]. Participants with a self-reported history of mental health problems were more afraid of COVID-19 and more worried about their loved ones contracting COVID-19, had higher distress, depression, anxiety, health anxiety and contamination fears, and higher rates of elevated health anxiety (26% versus 11%) than those without pre-existing mental health diagnoses. Relative to those without mental health issues, a greater proportion of people with self-reported mental health problems had elevated health anxiety (26% versus 11%), and said their mental health had been 'a lot worse' since the outbreak (26% versus 13%). Having a history of mental health issues predicted higher depression, anxiety and stress levels.

Because we did not collect any information about the history and nature of these mental health diagnoses, we cannot determine whether these individuals had higher distress prior to the pandemic, or whether distress increased as a result of the pandemic, due to inability to access usual supports, social isolation or loneliness [7]. However, our findings highlight the need for proactive mental health interventions for those who are experiencing elevated symptoms of depression, anxiety and stress during the current COVID-19 pandemic, regardless of whether the distress is an exacerbation or recurrence of pre-existing mental health concerns, or new onset. Digital interventions, which have been shown to be highly effective and cost-effective for depression and anxiety treatment [48] will be crucial to respond to these ongoing mental health concerns, as they have capacity to deliver high quality interventions for distress at scale, and to those in social isolation who are unable to attend face-to-face services [7, 49].

This study provides new knowledge about the rates of health anxiety during the COVID-19 pandemic. Over one in four (26%) of people with a prior history of mental health issues, and 11% of those without pre-existing mental health issues reported elevated health anxiety in the past week, which is higher than rates of health anxiety in the general Australian population (3.4%) [50], and closer to the rates of health anxiety observed in general practice (10%) and outpatient medical clinic settings (20–25%) [51]. While these symptoms are not necessarily indicative of illness anxiety disorder, high health anxiety is likely to have significant ramifications for health service utilisation. Responses to health anxiety vary substantially, ranging from a complete avoidance of doctors, hospitals, and medical settings due to fear, to the other end of the spectrum of excessive, repeated, and unnecessary health service use, diagnostic testing, emergency visits and paramedic calls [52]. Proactive treatment of health anxiety with digital interventions may also be needed should these symptoms persist [53, 54].

In prior research, risk perceptions, including the perceived risk of contracting the virus, perceived control over the virus, and the perceived seriousness of the symptoms have been shown to be associated with psychological distress, and behavioural responses to disease outbreaks. Consistent with the findings of SARS pandemics, and our previous study, we found moderate perceptions of risk of contracting the virus. Participants rated on average that there was a 50% likelihood of contracting the virus personally, and higher perceived risk was associated with higher depression and stress levels. In the current cohort approximately one third of participants expected COVID-19 to lead to severe symptoms (32.1%), and in some cases death (4%), which is higher than in our previous study, where we found only 25% expected severe symptoms. The expected severity of the COVID-19 illness differs markedly to the reality for most people, as studies show that 80% of people will experience no or mild symptoms [55]. These findings reinforce the need for education campaigns to address these misperceptions, especially as research has shown that these beliefs are associated with engagement with distress. These risk perceptions explained a relatively small amount of variance in the regression analyses, with perceived control over COVID-19 a consistent predictor of better mental health and higher perceived severity of illness associated with higher depression and anxiety. However, it

is important to note that other predictors, including loneliness, financial stress, uncertainty, demographic factors, and prior history of mental and chronic illness were stronger predictors of distress. These findings suggest that sociodemographic factors, and general uncertainty about the future, financial stress, and physical and mental health status may play a more important role in contributing to psychological distress during the COVID-19 pandemic than specific fears or perceived risk of contracting the virus. However, it is also possible that individuals with these risk factors were already distressed prior to the pandemic. The small variance explained by the COVID-19 variables might also be due to the fact that variance in distress was accounted for other variables (e.g., uncertainty about the future, hygiene behaviours) that were entered in earlier steps of the regression analyses. This finding requires replication, and future studies should examine whether the COVID-19 risk perception variables predict behavioural responses to the pandemic (e.g., hand-washing, social distancing), contamination fears, or health anxiety to a greater degree than general psychological distress.

Similar to Wang et al. [8], some of the most common precautionary behaviours were avoiding touching objects that had been touched by others, washing hands, and using hand sanitiser. Participants also commonly reported staying at home and avoiding social events and socialising with others outside of the household. In contrast to media portrayals of panic buying, excessive purchasing behaviour was not common. In previous research, higher engagement in hygiene behaviours, such as handwashing have been associated with lower distress and anxiety, suggesting behavioural control may be protective for mental health. However, in the current cohort we found some inconsistent results, with engagement in *more* hygiene behaviours associated with *higher* anxiety and stress levels (they were not associated with depression). These findings differ to the findings of Wang et al. [8] during the early stages of the epidemic in China, where the use of precautionary measures, such as avoiding sharing utensils, hand hygiene and wearing masks were associated with lower stress, anxiety and depression. However, the current findings are consistent with some research from the SARS epidemic, in which moderate levels of anxiety were associated with higher uptake of precautionary behaviours [56]. It is possible that the association we found was due to people who were higher in anxiety or stress using these behaviours in an attempt to control anxiety.

Finally, concerns have been raised about the potential impact of social isolation and quarantine on physical inactivity, as well as increased alcohol use and abuse. On the AUDIT-C brief screener for alcohol use, approximately 52.7% met criteria for hazardous drinking levels, which is higher than the 42% found in primary care samples in Australia [57] and higher than USA-based population samples (35%-45%) [58]. However, people in self-isolation reported lower alcohol consumption than those who were not self-isolating. It is also important to note that participants with a prior experience of mental health problems had lower rates of hazardous drinking, and lower rates of inactivity. In the current sample, 42.7% met the national physical activity recommendations of 150 minutes or more of moderate to vigorous activity over the past week, which are similar to the population based normative data from the Australian National Health survey (43–44%) [47]. We will be following up these participants longitudinally to explore whether activity levels decrease further as isolation restrictions proceed. Given the importance of exercise and physical activity in maintaining mental health and promoting overall health and wellbeing, interventions could be used to assist increasing activity levels for those sedentary at home.

## Limitations

The results are based on a convenience sample recruited online, who were mostly women (85%) and well educated, and a significant proportion reported having lived experience of a

mental health diagnosis (70%). This may overestimate the symptom severity and impact of COVID-19, especially given past studies have shown worse impact of pandemics on those with pre-existing mental illness, and in females. It may also mean that the results cannot generalise to the broader Australian population. The rate of participants who had family and/or friends who had contracted COVID-19 (4.8%) was higher than expected based on the low prevalence of COVID-19 in the community, which also may not be generalise to the broader community. Results are also based solely on validated self-report measures, due to their ease and speed of assessment, and administration, and we did not independently check whether survey respondents had IP addresses originating in Australia. Conducting diagnostic interviews to assess mental health diagnoses with more than 5000 participants in 10 days would not have been feasible. Future studies need to explore the impact of COVID-19 on mental health of COVID-19 patients, given evidence of increased rates of Post -Traumatic Stress Disorder, sleep disturbance and depression in SARS patients [5, 59]. Finally, the study was cross-sectional; the next step in our research is to track this cohort over time, to explore how their mental health changes as the pandemic evolves in Australia.

## Supporting information

**S1 File. Advertisement used for recruitment on Facebook, LinkedIn and Twitter.**
(DOCX)

**S1 Table. Mental health in people with and without a prior self-reported mental health diagnosis.**
(DOCX)

**S2 Table. Comparison between those in self-isolation versus not in self isolation.**
(DOCX)

## Acknowledgments

We thank all of the participants who kindly contributed to this study.

## Author Contributions

**Conceptualization:** Jill M. Newby, Kathleen O'Moore, Samantha Tang, Kate Faasse.

**Formal analysis:** Jill M. Newby, Kate Faasse.

**Funding acquisition:** Jill M. Newby.

**Investigation:** Jill M. Newby.

**Writing – original draft:** Jill M. Newby, Kathleen O'Moore, Samantha Tang, Kate Faasse.

**Writing – review & editing:** Jill M. Newby, Kathleen O'Moore, Samantha Tang, Helen Christensen, Kate Faasse.

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
