## [Decision Letter · Decision Letter 0]

16 Jun 2020

PONE-D-20-12728

Acute mental health responses during the COVID-19 pandemic in Australia

PLOS ONE

Dear Dr. Tang,

Thank you for submitting your manuscript to PLOS ONE. After careful consideration, we feel that it has merit but does not fully meet PLOS ONE’s publication criteria as it currently stands. Therefore, we invite you to submit a revised version of the manuscript that addresses the points raised during the review process.

We look forward to receiving your revised manuscript.

Kind regards,

Joel Msafiri Francis, MD, MS, PhD

Academic Editor

PLOS ONE

Journal Requirements:

Reviewers' comments:

Reviewer's Responses to Questions

**Comments to the Author**

1. Is the manuscript technically sound, and do the data support the conclusions?

Reviewer #1: Yes

Reviewer #2: Yes

Reviewer #3: Partly

2. Has the statistical analysis been performed appropriately and rigorously? 

Reviewer #1: No

Reviewer #2: Yes

Reviewer #3: I Don't Know

3. Have the authors made all data underlying the findings in their manuscript fully available?

Reviewer #1: No

Reviewer #2: Yes

Reviewer #3: No

4. Is the manuscript presented in an intelligible fashion and written in standard English?

Reviewer #1: Yes

Reviewer #2: Yes

Reviewer #3: Yes

5. Review Comments to the Author

Reviewer #1: Overall, a well written and timely manuscript on an important topic. Main study limitations (e.g., limited sample representativeness) were identified and acknowledged. Findings of the exploratory analyses are of interest to a broad range of audiences. My main concern relates to the lack of information about the analytical procedures in the Methods section. Further comments and suggestions to strengthen the manuscript are as follows:

Introduction

- Further information about the COVID-19 containment regime and restrictions in place in Australia at the time (e.g., self-isolation, social distancing measures) could be provided (beyond the mandatory quarantine for overseas travelers) to better contextualise the study setting for readers.

Methods

- The Methods section does not appear to include any information on the types of ‘data analyses’ conducted (e.g., descriptive analyses, Independent samples t test, chi square analysis, linear regressions), which would be important to incorporate.

Results

- (p.9, lines 213-14): suggest reordering to read: “The majority of participants rated their health as ‘good’ (37.7%), ‘very good’ (24.4%) or ‘fair’ (24.4%), …”

- (Table 1): decimal places for reported percentages could be kept consistent; for the Table to work standalone, it could include a footnote indicating the varied time-frames that mental health and general health measures referred to (e.g., over the past week, or lifetime)

- (Mental Health): The fact that more than 70% of respondents self-reported a lifetime mental health diagnosis strikes me as comparatively high and should be mentioned upfront in text when reporting sample descriptives. Likewise, 45% of respondents receiving current mental health treatment does seem equally elevated compared to general population stats and would also be worth reporting in text (as both have implications for sample representativeness).

- (Table 4): row labels in the first column are difficult to read and could be reformatted

- (p.20, line 299): please check finding – as ‘being unemployed’ does not seem to be significantly associated with depression (compare Table 8)

Discussion

- (p.27, lines 443-448): while overarching AUDIT-C scores for this sample are indeed concerning, it could also be worth mentioning that those in self-isolation reported lesser alcohol consumption than those not self-isolating (compare Table 5)

- The striking finding that COVID variables accounted for very little variance in DASS scores would be worthy of further interpretation and discussion (e.g., in terms of potential methodological or conceptual implications)

Minor Typos

- (p.5, line 116): identify vulnerable groups who are [at] risk

- (p.7, line 153): hea[l]th

- (p.8, line 176): were administered in randomised [in] order to

- (p.26, line 416): was associate[d]

Reviewer #2: On the whole, this is a well-designed study of the psychological impact of the COVID-19 pandemic in Australia. The study authors have clearly made an attempt to extend the work of earlier authors, and to improve on it in some aspects.

That being said, the following concerns still exist:

1. Apart from the Chinese studies cited, a few studies from outside China have been published. It would be worth discussing these papers briefly in the Introduction or Discussion sections, to provide a broader context for the authors' work.

2. A measure of compulsive washing was included in the study. How did it correlate with measures of anxiety, depression or stress?

3. Given the small number of gender non-binary and Aboriginal / Islander subjects in the study, it may be premature to label these factors as robust predictors of mental health outcomes.

4. For the purpose of reading ease, it may be preferable to present only the key findings in tables and provide the remainder of the information as supplementary material.

Reviewer #3: The manuscript is well written. It addresses an important contemporary issue and provides information no available elsewhere in the literature. The key limitation in this manuscript is the sampling approach and the lack of a description and justification of the analysis plan.

* While the authors do address the sampling issues in their limitations briefly, I think that acknowledging this in more detail and presenting the findings in a more tentative manner would improve the publication.

* The planned analyses are not described in the Methods. Additionally, inadequate justification is provided about why a step-wise approach to linear regression was adopted, and the rationale underlying the sequence and variables included at each step.

Methods > Recruitment

* Given that the sample that emerged does not appear representative of the Australian population generally it would be useful to include the survey advertisement as a supplementary material. The phrasing of the invitation to participate would foreseeably impact the sample composition.

* Unless IP addresses external to Australia were blocked from survey participation it would be useful to describe in the results confirmation that all included IP addresses originated in Australia. Also, were any duplicate IP addresses identified? How were these managed?

Methods>Participants

* Was any analysis completed comparing the completers versus the non-completers? It appears that any incomplete surveys were omitted from the data-set... how might this have biased the findings?

* Is the 15-minute completion timeframe based on actual participants or the estimation of the researchers? This is a relatively long survey and I am surprised by the completion rate. I wonder why this was so high - might it tell something about the sampling approach?

* Did clicking on the link in social media open the PICF? Or did it open another page where potential participants needed to click to open the PICF? If the former it would be really helpful to include information about how many people clicked on the link without subsequently accessing the PICF.

Methods > ****

Nil Analyses section is provided. It would be helpful to describe how the data was intended to be analysed a priori, and in particular the rationale for the step-wise approach to the regression.

Results>Demographics

* The proportion of participants who identified as female gender and self-reported previous mental illness suggests that the sample is not representative. While this is discussed in the limitations I think it would be useful to consider this in greater detail in the Discussion. Why did the sampling approach result in such a high proportion of educated women with pre-existing mental illness participating? Similarly it suggests the relevance of more tentative language in the Discussion.

Results>Health related information

* 4.8% of the sample reported having a friend of family member who had contracted COVID-19. Given the very low prevalence of this condition in Australia this again suggests an issue with the sampling that needs to be addressed in the limitations.

Results>Mental Health

* Symbol for statistical test at Line 251 is missing, presumably Chi squared

Results > Table 3

* Consider including the median for the DASS scores to given a clearer picture of central tendency

Results > Table 5

You have combined those in voluntary and required self-isolation. Were any analyses completed to see that the patter of results still held when the required self-isolation were excluded? Higher distress and disconnection would be foreseeable for this group.

Discussion:

See general comments about the need to discuss sampling issues in more detail and present findings more tentatively.

6. PLOS authors have the option to publish the peer review history of their article (what does this mean?). If published, this will include your full peer review and any attached files.

Reviewer #1: No

Reviewer #2: Yes: Ravi Philip Rajkumar

Reviewer #3: Yes: Dr Stephen Parker

---

## [Author Response · Author response to Decision Letter 0]

1 Jul 2020

Please see 'Response to reviewers' file attachment.

---

## [Decision Letter · Decision Letter 1]

10 Jul 2020

Acute mental health responses during the COVID-19 pandemic in Australia

PONE-D-20-12728R1

Dear Dr. Newby,

We’re pleased to inform you that your manuscript has been judged scientifically suitable for publication and will be formally accepted for publication once it meets all outstanding technical requirements.

Kind regards,

Joel Msafiri Francis, MD, MS, PhD

Academic Editor

PLOS ONE

Additional Editor Comments (optional):

Reviewers' comments:

Reviewer's Responses to Questions

**Comments to the Author**

1. If the authors have adequately addressed your comments raised in a previous round of review and you feel that this manuscript is now acceptable for publication, you may indicate that here to bypass the “Comments to the Author” section, enter your conflict of interest statement in the “Confidential to Editor” section, and submit your "Accept" recommendation.

Reviewer #1: All comments have been addressed

Reviewer #2: All comments have been addressed

Reviewer #3: All comments have been addressed

2. Is the manuscript technically sound, and do the data support the conclusions?

Reviewer #1: Yes

Reviewer #2: Yes

Reviewer #3: Yes

3. Has the statistical analysis been performed appropriately and rigorously? 

Reviewer #1: Yes

Reviewer #2: Yes

Reviewer #3: Yes

4. Have the authors made all data underlying the findings in their manuscript fully available?

Reviewer #1: No

Reviewer #2: Yes

Reviewer #3: No

5. Is the manuscript presented in an intelligible fashion and written in standard English?

Reviewer #1: Yes

Reviewer #2: Yes

Reviewer #3: Yes

6. Review Comments to the Author

Reviewer #1: The authors have satisfactorily addressed all previous comments raised by this reviewer. The manuscript has gained in clarity and been strengthened as a consequence.

Reviewer #2: (No Response)

Reviewer #3: Thank you for addressing the concerns that were raised in my initial review. This article is timely and addresses an important issue. The methodological limitations are now adequately acknowledged.

7. PLOS authors have the option to publish the peer review history of their article (what does this mean?). If published, this will include your full peer review and any attached files.

Reviewer #1: No

Reviewer #2: **Yes: **Ravi Philip Rajkumar

Reviewer #3: **Yes: **Stephen Parker

---

## [Editor Report · Acceptance letter]

17 Jul 2020

PONE-D-20-12728R1 

Acute mental health responses during the COVID-19 pandemic in Australia 

Dear Dr. Newby:

I'm pleased to inform you that your manuscript has been deemed suitable for publication in PLOS ONE. Congratulations! Your manuscript is now with our production department. 

Kind regards, 

on behalf of

Dr. Joel Msafiri Francis 

Academic Editor

PLOS ONE